# Invertibility of Convolutional Generative Networks from Partial Measurements

**Fangchang Ma***
MIT
fcma@mit.edu

**Ulas Ayaz***
MIT
uayaz@mit.edu
uayaz@lyft.com

**Sertac Karaman**
MIT
sertac@mit.edu

## Abstract

The problem of inverting generative neural networks (*i.e.*, to recover the input latent code given partial network output), motivated by image inpainting, has recently been studied by a prior work that focused on fully-connected networks. In this work, we present new theoretical results on convolutional networks, which are more widely used in practice. The network inversion problem is highly non-convex, and hence is typically computationally intractable and without optimality guarantees. However, we rigorously prove that, for a 2-layer convolutional generative network with ReLU and Gaussian-distributed random weights, the input latent code can be deduced from the network output efficiently using simple gradient descent. This new theoretical finding implies that the mapping from the low-dimensional latent space to the high-dimensional image space is one-to-one, under our assumptions. In addition, the same conclusion holds even when the network output is only partially observed (*i.e.*, with missing pixels). We further demonstrate, empirically, that the same conclusion extends to networks with multiple layers, other activation functions (leaky ReLU, sigmoid and tanh), and weights trained on real datasets.

## 1 Introduction

In recent years, generative models have made significant progress in learning representations for complex and multi-modal data distributions, such as those of natural images [10, 18]. However, despite the empirical success, there has been relatively little theoretical understanding into the mapping itself from the input latent space to the high-dimensional space. In this work, we address the following question: given a convolutional generative network[2], is it possible to "decode" an output image and recover the corresponding input latent code? In other words, we are interested in the *invertibility of convolutional generative models*.

The impact of the network inversion problem is two-fold. Firstly, the inversion itself can be applied in image in-painting [21, 17], image reconstruction from sparse measurements [14, 13], and image manipulation [22] (*e.g.*, vector arithmetic of face images [12]). Secondly, the study of network inversion provides insight into the mapping from the low-dimensional latent space to the high-dimensional image space (*e.g.*, is the mapping one-to-one or many-to-one?). A deeper understanding of the mapping can potentially help solve the well known mode collapse[3] problem [20] during the training in the generative adversarial network (GAN) [7, 16].

---

[*]Both authors contributed equally to this work. Ulas Ayaz is presently affiliated with Lyft, Inc.

[2]Deep generative models typically use transposed convolution (*a.k.a.* "deconvolution"). With a slight abuse of notation we refer to transposed convolutional generative models as convolutional models.

[3]Mode collapse refers to the problem that the Generator characterizes only a few images to fool the discriminator in GAN. In other words, multiple latent codes are mapped to the same output in the image space.

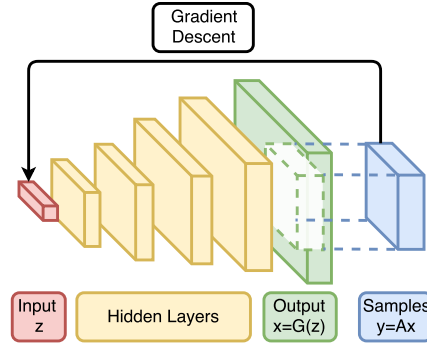

Figure 1: Recovery of the input latent code $z$ from under-sampled measurements $y = AG(z)$ where $A$ is a sub-sampling matrix and $G$ is an expanding generative neural network. We prove that $z$ can be recovered with guarantees using simple gradient-descent methods under mild technical assumptions.

The challenge of the inversion of a deep neural network lies in the fact that the inversion problem is highly non-convex, and thus is typically computationally intractable and without optimality guarantees. However, in this work, we show that network inversion can be solved efficiently and optimally, despite being highly non-convex. Specifically, we prove that with simple first-order algorithms like stochastic gradient descent, we can recover the latent code with guarantees. The sample code is available at `https://github.com/fangchangma/invert-generative-networks`.

## 1.1 Related Work

The network inversion problem has attracted some attention very recently. For instance, Bora et al. [2] empirically find that minimizing the non-convex Problem (3), which is defined formally in Section 2, using standard gradient-based optimizer yields good reconstruction results from small number of Gaussian random measurements. They also provide guarantees on the global minimum of a generative network with certain structure. However, their work does not analyze how to find the global minimum. Hand and Voroninski [8] further establish that a fully connected generative network with weights following Gaussian distribution can be inverted given only compressive linear observations of its last layer. In particular, they show that under mild technical conditions Problem (3) has a favorable global geometry, in the sense that there are no stationary points outside of neighborhoods around the desired solution and its negative multiple with high probability. However, most practical generative networks are deconvolutional rather than fully connected, due to memory and speed constraints. Besides, their results are proved for Gaussian random measurements, which are rarely encountered in practical applications. In this work, we build on top of [8] and extend their results to 2-layer deconvolutional neural networks, as well as uniform random sub-sampling.

We also note the work [6], which studies a 1-layer network with a special activation function (Concatenated ReLU, which is essentially linear) and a strong assumption on the latent code ($k$-sparsity). In comparison, our results are much stronger than [6]. Specifically, our results are for 2-layer networks (with empirical evidences for deeper networks), and they apply to the most common ReLU activation function. Our result also makes no assumption regarding the sparsity of latent codes.

Another line of research, which focuses on gradient-based algorithms, analyzes the behavior of (stochastic) gradient descent for Gaussian-distributed input. Soltanolkotabi [19] showed that projected gradient descent is able to find the true weight vector for 1-layer, 1-neuron model. More recently, Du et al. [5] improved this result for a simple convolutional neural network with two unknown layers. Their assumptions on random input and their problem of weight learning are different than the problem we study in this paper.

Our problem is also connected to *compressive sensing* [4, 3] which exploits the sparsity of natural signals to design acquisition schemes where the number of measurements scales linearly with the sparsity level. The signal is typically assumed to be sparse in a given dictionary, and the objective function is convex. In comparison, our work does not assume sparsity, and we provide a direct analysis of gradient descents for the highly non-convex problem.

## 1.2 Contribution

The contribution of this work is three-fold:

- We prove that a convolutional generative neural network is invertible, with high probability, under the following assumptions: (1) the network consists of two layers of transposed convolutions followed by ReLU activation functions; (2) the network is (sufficiently) expansive; (3) the filter weights follow a Gaussian distribution. When these conditions are satisfied, the input latent code can be recovered from partial output of a generative neural network by minimizing a $\mathcal{L}_2$ empirical loss function using gradient descent.
- We prove that the same inversion can be achieved with high probability, even when only a subset of pixels is observed. This is essentially the image inpainting problem.
- We validate our theoretical results using both random weights and weights trained on real data. We further demonstrate empirically that that our theoretical results generalize to (1) multiple-layer networks; (2) networks with other nonlinear activation functions, including Leaky ReLU, Sigmoid and Tanh.

Two key ideas of our proof include (a) the concentration bounds of convolutional weight matrices combined with ReLU operation, and (b) the angle distortion between two arbitrary input vectors under the transposed convolution and ReLU. In general, our proof follows a similar basic structure to [8], where the authors show the invertibility of fully connected networks with Gaussian weights. However, in fully connected networks, the weight matrix of each layer is a dense Gaussian matrix. In contrast, in convolutional networks the weight matrices are highly sparse with block structure due to striding filters, as in Figure 2(a). Therefore, [8]'s proof does not apply to convolutional networks, and the extension of concentration bounds for our case is not trivial.

To address such problem, we propose a new permutation technique which shuffles the rows and columns of weight matrices to obtain a block matrix, as depicted in Figure 2(b). With permutation, each block is now a dense Gaussian matrix, where we can apply existing matrix concentration results. However, the permutation operation is quite arbitrary, depending on the structure of the convolutional network. This requires some careful handling, since the second step (b) requires the control of angles.

In addition, Hand and Voroninski [8] assume a Gaussian sub-sampling matrix at the output of the network, rather than partial sub-sampling (sub-matrix of identity matrix) that we study in this problem. We observe that sub-sampling operation can be swapped with the last ReLU in the network, since both are entrywise operations. We handle the sub-sampling by making the last layer more expansive, and prove that it is the same with no downsampling from a theoretical standpoint.

## 2 Problem Statement

In this section, we introduce the notation and define the network inversion problem. Let $z^\diamond \in \mathbb{R}^{n_0}$ denote the latent code of interest, $G(\cdot) : \mathbb{R}^{n_0} \to \mathbb{R}^{n_d}$ ($n_0 \ll n_d$) be a $d$-layer generative network that maps from the latent space to the image space. Then the ground truth output image $x^\diamond \in \mathbb{R}^{n_d}$ is produced by

$$x^\diamond = G(z^\diamond), \tag{1}$$

In this paper we consider $G(\cdot)$ to be a deep neural network[4]. In particular we assume $G(\cdot)$ to be a two-layer transposed convolutional network, modeled by

$$G(z) = \sigma(W_2 \sigma(W_1 z)) \tag{2}$$

where $\sigma(z) = \max(z, 0)$ denotes the rectified linear unit (ReLU) that applies entrywise. $W_1 \in \mathbb{R}^{n_1 \times n_0}$ and $W_2 \in \mathbb{R}^{n_2 \times n_1}$, are the weight matrices of the convolutional neural network in the first and second layers, respectively. Note that since $G(\cdot)$ is a convolutional network, $W_1$ and $W_2$ are highly sparse with a particular block structure, as illustrated in Figure 2(a).

Let us make the inversion problem a bit more general by assuming that we only have partial observations of the output image pixels. Specifically, let $A \in \mathbb{R}^{m \times n_2}$ be a sub-sampling matrix

(a subset of the rows of an identity matrix), and then the observed pixels are $y^\diamond = Ax^\diamond \in \mathbb{R}^m$. Consequently, the inversion problem given partial measurements can be described as follows:

Let: $z^\diamond \in \mathbb{R}^{n_0}, W_1 \in \mathbb{R}^{n_1 \times n_0}, W_2 \in \mathbb{R}^{n_2 \times n_1}, A \in \mathbb{R}^{m \times n_2}$

Given: $A, W_1, W_2$ and observations $y^\diamond = AG(z^\diamond)$

Find: $z^\diamond$ and $x^\diamond = G(z^\diamond)$

Since $x^\diamond$ is determined completely by the latent representation $z^\diamond$, we only need to find $z^\diamond$. We propose to solve the following optimization problem for an estimate $\hat{z}$:

$$\hat{z} = \arg\min_z J(z), \text{ where } J(z) = \frac{1}{2} \|y^\diamond - AG(z)\|^2 \qquad (3)$$

This minimization problem is highly non-convex because of $G$. Therefore, in general a gradient descent approach is not guaranteed to find the global minimum $z^\diamond$, where $J(z^\diamond) = 0$.

## 2.1 Notation and Assumptions

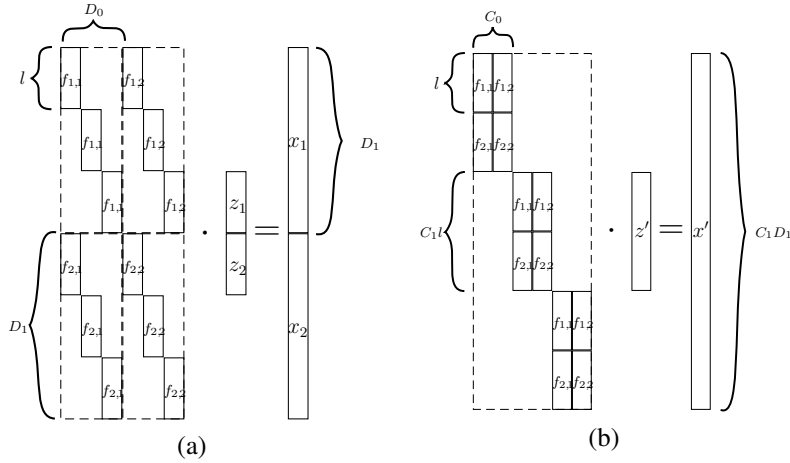

(a)                    (b)

Figure 2: Illustration of a single transposed convolution operation. $f_{i,j}$ stands for $i^{th}$ filter kernel for the $j^{th}$ input channel. $z$ and $x$ denote the input and output signals, respectively. (a) The standard transposed convolution represented as linear multiplication. (b) With proper row and column permutations, the permuted weight matrix has a repeating block structure.

We vectorize the input signal to 1D signal. The feature at the $i^{th}$ layer consists of $C_i$ channels, each of size $D_i$. Therefore, $n_i = C_i \cdot D_i$. At any convolutional layer, let $f_{i,j}$ denotes the kernel filter (each of size $\ell$) for the $i^{th}$ input channel and the $j^{th}$ output channel. For simplicity, we assume the stride to be equal to the kernel size $l$. All filters can be concatenated to form a large block matrix $W_i$. For instance, an example of such block matrix $W_1$ for the first layer is shown in Figure 2(a). Under our assumptions, the input and output sizes at each deconvolution operation can be associated as $D_{i+1} = D_i \ell$.

Let $D_v J(x)$ be one-sided directional derivative of the objective function $J(\cdot)$ along the direction $v$, *i.e.*, $D_v J(x) = \lim_{t \to 0^+} \frac{J(x+tv) - J(x)}{t}$. Let $\mathcal{B}(x, r)$ be the Euclidean ball of radius $r$ centered at $x$. We omit some universal constants in the inequalities and use $\gtrsim_\epsilon$ (if the constant depends on a variable $\epsilon$) instead.

## 3 Main Results

In this section, we present our main theoretical results regarding the invertibility of a 2-layer convolutional generative network with ReLUs. Our first main theoretical contribution is as follows: although the problem in (3) is non-convex, under appropriate conditions there is a strict descent direction everywhere, except in the neighborhood of $z^\diamond$ and that of a negative multiple of $z^\diamond$.

**Theorem 1** (Invertibity of convolutional generative networks). *Fix $\epsilon > 0$. Let $W_1 \in \mathbb{R}^{C_0 D_0 \times C_1 D_1}$ and $W_2 \in \mathbb{R}^{C_1 D_1 \times C_2 D_2}$ be deconvolutional weight matrices with filters in $\mathbb{R}^{\ell}$ with i.i.d. entries from $\mathcal{N}(0, 1/C_i\ell)$ for layers $i = 1, 2$ respectively. Let the sampling matrix $A = I$ be an identity matrix (meaning there's no sub-sampling). If $C_1\ell \gtrsim_\epsilon C_0 \log C_0$ and $C_2\ell \gtrsim_\epsilon C_1 \log C_1$ then with probability at least $1 - \kappa(D_1 C_1 \, e^{-\gamma C_0} + D_2 C_2 \, e^{-\gamma C_1})$ we have the following. For all nonzero $z$ and $z^\diamond$, there exists $v_{z,z^\diamond} \in \mathbb{R}^{n_0}$ such that*

$$D_{v_{z,z^\diamond}} J(z) < 0, \qquad \forall z \notin \mathcal{B}(z^\diamond, \epsilon\|z^\diamond\|_2) \cup \mathcal{B}(-\rho z^\diamond, \epsilon\|z^\diamond\|_2) \cup \{0\} \qquad (4)$$

$$D_z J(0) < 0, \qquad \forall z \neq 0, \qquad (5)$$

*where $\rho$ is a positive constant. Both $\gamma > 0$ and $\kappa > 0$ depend only on $\epsilon$.*

Theorem 1 establishes under some conditions that the landscape of the cost function is not adversarial. Despite the heavily loaded notation, Theorem 1 simply requires that the weight matrices with Gaussian filters should be sufficiently expansive (*i.e.*, output dimension of each layer should increase by at least a logarithmic factor). Theorem 1 does not provide information regarding the neighborhood centered at $-\rho x^\diamond$, which implies the possible existence of a local minimum or a saddle point. However, empirically we did not observe convergence to a point other than the ground truth. In other words, gradient descent seems to always find the global minimum, see Figure 4.

One assumption we make is the size of stride $s$ being same as the filter size $\ell$. Although theoretically convenient, this assumption is not common in the practical choices of transposed convolutional networks. We believe a further analysis can remove this assumption, which we also leave as a future work. In practice different activation functions other than ReLU can be used as well, such as sigmoid function, Tanh and Leaky ReLU. It is also an interesting venue of research to see whether a similar analysis can be done with those activations. In particular, for Leaky ReLU we briefly explain how the proof would divert from ours in Section Sup.2. We include landscapes of the cost function when different activations are used in Figure 4.

Gaussian weight assumption might seem unrealistic at first. However, there is some research [1] indicating that weights of some trained networks follow a normal distribution. We also make a similar observation on the networks we trained, see Section 4. We also note that Theorem 1 does not require independence of network weights across layers.

**Proof Outline:** Due to space limitations, the complete proof of Theorem 1 is given in the supplementary material [15]. Here we give a brief outline of the proof and highlight the main steps. The theorem is proven by showing two main conditions on the weight matrices.

The first condition is on the spatial arrangement of the network weights within each layer. Lemma Sup.2 [15] provides a concentration bound on the distribution of the effective weight matrices (after merging the ReLUs into the matrices). It shows that the set of neuron weights within each layer are distributed approximately like Gaussian. A key idea for the proving Lemma Sup.2 is our new *permutation* technique. Specifically, we rearrange both rows and columns of the sparse weight matrices, as in Figure 2(a), into a block diagonal matrix, as in Figure 2(b). Each block in the permuted matrix is the same Gaussian matrix with independent entries. The permutation into block matrices helps turns each block in Figure 2(b) into a dense Gaussian matrix, and therefore makes it possible to utilize existing concentration bounds on Gaussian matrices.

The second condition is on the approximate angle contraction property of an effective weight matrix $W_i$ (after merging the ReLUs into the matrices). Lemma Sup.4 [15] shows that the angle between two arbitrary input vectors $x$ and $y$ does not vanish under a transposed convolution layer and the ReLU. The permutation poses a significant challenge on the proof of Lemma Sup.4, since permutation of the input vectors distorts the angles. The difficulty is handled carefully in the proof of Lemma Sup.4, which deviates from the proof machinery in [8] and hence is a major technical contribution. □

**Corollary 2** (One-to-one mapping). *Under the assumptions of Theorem 1, the mapping $G(\cdot) : \mathbb{R}^{n_0} \to \mathbb{R}^{n_2}$ ($n_0 \ll n_2$) is injective (i.e., one-to-one) with high probability.*

Corollary 2 is a direct implication of Theorem 1. Corollary 2 states that the mapping from the latent code space to the high-dimensional image space is one-to-one with high probability, when the assumptions hold. This is interesting from a practical point of view, because mode collapse is a well-known problem in training of GAN [20] and Corollary 2 provides a sufficient condition to avoid mode collapses. It remains to be further explored how we can make use of this insight in practice.

**Conjecture 3.** *Under the assumptions of Theorem 1, let the network weights follow any zero-mean subgaussian distribution $\mathbb{P}(|x| > t) \leqslant ce^{-\gamma t^2}$, $\forall t > 0$ instead of Gaussian. Then with high probability the same conclusion holds.*

A subgaussian distribution (*a.k.a.* light-tailed distribution) is one whose tail decays at least as fast as a Gaussian distribution (*i.e.*, exponential decay). This includes, for example, any bounded distribution and the exponential distribution. Empirically, we observe that Theorem 1 holds for a number of zero-mean subgaussian distributions, including uniform random weights and $\{+1, -1\}$ binary random weights.

Now let us move on to the case where the subsampling matrix $A$ is not an identity matrix. Instead, consider a fixed sampling rate $r \in (0, 1]$.

**Theorem 4** (Invertibility under partial measurements). *Under the assumptions of Theorem 1, let $A \in \mathbb{R}^{m \times C_2 D_2}$ be an arbitrary subsampling matrix with $m/(C_2 D_2) \geqslant r$. Then with high probability the same result as Theorem 1 hold.*

Note that the subsampling rate $r$ appears in the dimension of the weight matrix of the second layer.

*Proof.* Since ReLU operation is pointwise, we have the identity

$$y = AG(z) = A\sigma(W_2\sigma(W_1 z)) = \sigma(AW_2\sigma(W_1 z)).$$

It suffices to show that Theorem 1 still holds with $AW_2$ as the last weight matrix. Note that $AW_2$ selects a row subset of the matrix $W_2$ Figure 2(a). Consequently, after proper permutation, $AW_2$ is again a block diagonal matrix with each block being a Gaussian matrix with independent entries. Only this time the blocks are not identical, but instead have different sizes. As a result, Theorem 1 still holds for $AW_2$, since the proof of Theorem 1 does not require the identical blocks. However, there are certain dimension constraints, which can be met by expanding the last layer with a factor of $r$, the sampling rate. This modification is reflected in the additional dimension assumption on the weight matrix $W_2$. □

The minimal sampling rate $r$ is a constant that depends on both the network architecture (e.g., how expansive the networks are) and the sampling matrix $A$. We made 2 empirical observations. Firstly, spatially disperse sampling patterns (e.g., uniform random samples) require a lower $r$, whilst more aggressive sampling patterns (e.g., top half, left half, sampling around image boundaries) demand more measurements for perfect recovery. Secondly, regardless of the sampling patterns $A$, the probability of perfect recovery exhibits a phase transition phenomenon w.r.t. the sampling rate $r$. This observation supports Theorem 4 (i.e., network is invertible given sufficient measurements). A more rigorous and mathematical characterization of $r$ remains an open question.

## 4 Experimental Validation

In this section, we verify the gaussian weight assumption of trained generative networks, our main result Theorem 4 on simulated 2-layer networks, as well as the generalization of Theorem 4 to more complex multi-layer networks trained on real datasets.

### 4.1 Gaussian Weight in Trained Networks

We extract the convolutional filter weights, trained on real data to generate images in Figure 5, from a 4-layer convolutional generative models. The histogram of the weights in each layer is depicted in Figure 3. It can be observed that the trained weights highly resembles a zero-mean gaussian distribution. We also discover similar distributions of weights in other trained convolutional networks, such as ResNet [9]. Arora et al. [1] also report similar results.

### 4.2 On 2-layer Networks with Random Weights

As a sanity check on Theorem 4, we construct a generative neural network with 2 transposed convolution layers, each followed by a ReLU. The first layer has 16 channels and the second layer has 1 single channel. Both layers have a kernel size of 5 and a stride of 3. In order to be able to visualize

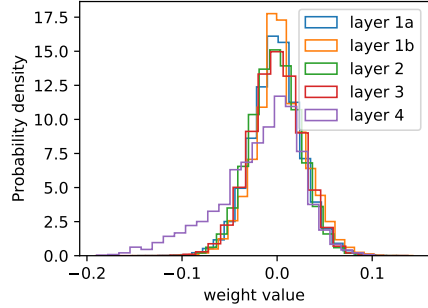

Figure 3: Distribution of the kernel weights from every layer in a trained convolutional generative network. The trained weights roughly follow a zero-mean gaussian distribution.

the cost function landscape, we set the input latent space to be 2-dimensional. The weights of the transposed convolution kernels are drawn i.i.d. from a Gaussian distribution with zero mean and unit standard deviation. Only 50% of the network output is observed. We compute the cost function $J(z)$ for every input latent code $z$ on a grid centered at the ground truth. The landscape of the cost function $J(z)$ is depicted in Figure 4(a). Although Theorem 4 implies a possibility of a stationary point at the negative multiple of the ground truth, experimentally we do not observe convergence to any point other than the global minimum.

Despite the fact that Theorem 1 and Theorem 4 are proved only for the case of 2-layer network with ReLU, the same conclusion empirically extends to networks with more layers and different kernel sizes and strides. In addition, the inversion of generative models generalizes to other standard activation functions including Sigmoid, and Tanh. Specifically, Sigmoid and Tanh have quasi-convex landscapes as shown in Figure 4(b) and (c), which are even more favorable than that of ReLU. Leaky ReLU has the same landscape as a regular ReLU.

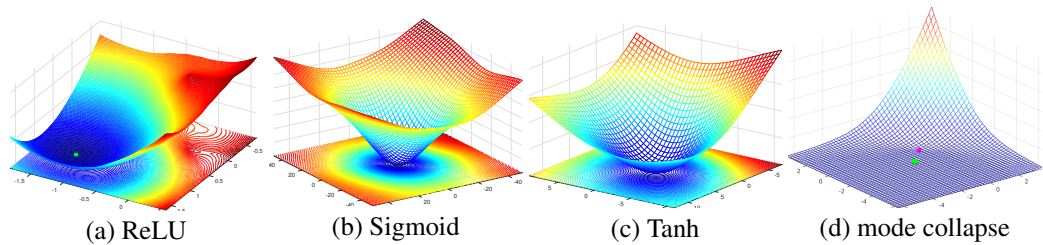

(a) ReLU    (b) Sigmoid    (c) Tanh    (d) mode collapse

Figure 4: The landscape of the cost function $J(z)$ for deconvolutional networks with (a) ReLU, (b) Sigmoid, and (c) Tanh as activation functions, respectively. There exists a unique global minimum.

As a counter example, we draw kernel weights uniformly randomly from $[0, 1]$ (which violates the zero-mean Gaussian assumption). Consequently, there is a flat global minimum in the latent space, as shown in Figure 4(d). In this region, any two latent vectors are mapped to the exact same output, indicating that mode collapse indeed occurs.

## 4.3 On Multi-layer Networks Trained with Real Data

In this section, we demonstrate empirically that our finding holds for multi-layer networks trained on real data. The first network is trained with GAN to generate handwritten digits, and the second for celebrity faces. In both experiments, the correct latent codes can be recovered perfectly from partial (but sufficiently many) observations.

**MNIST:**    For the first network on handwritten digit, we rescale the raw grayscale images from the MNIST dataset [11] to size of $32 \times 32$. We used the conditional deep convolutional generative adversarial networks (DCGAN) framework [16, 18] to train both a generative model and a discriminator. Specifically, the generative network has 4 transposed convolutional layers. The first 3 transposed convolutional layers are followed by a batch normalization and a Leaky ReLU. The last layer is followed by a Tanh. The discriminator has 4 convolutional layers, with the first 3 followed by batch

normalization and Leaky ReLU and the last one followed by a Sigmoid function. We use Adam with learn rate $0.1$ to optimize the latent code $z$. The optimization process usually converges within 500 iterations. The input noise to the generator is set to have a relatively small dimension 10 to ensure a sufficiently expanding network.

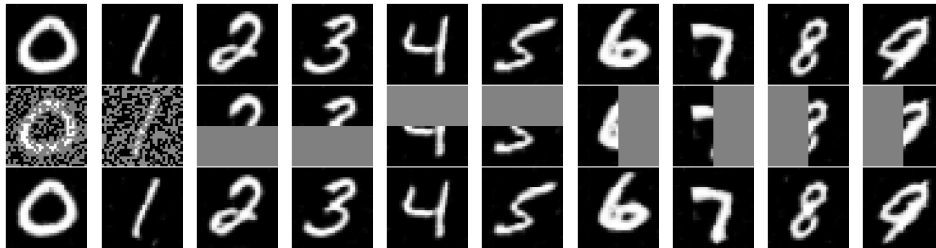

Figure 5: We demonstrate recovery of latent codes on a generative network trained on the MNIST dataset. From top to bottom: ground truth output images; partial measurements with different sampling masks; reconstructed image using the recovered latent codes from partial measurements. The recovery of latent codes in these examples is perfect, using simple gradient descent.

5 different sampling matrices are showcased in Figure 5, including observing uniform random samples, as well as the top half, bottom half, left half, and right half of the image space. In all cases, the input latent codes are recovery perfectly. We feed the recovered latent code as input to the network to obtain the completed image, shown in the $3^{rd}$ row.

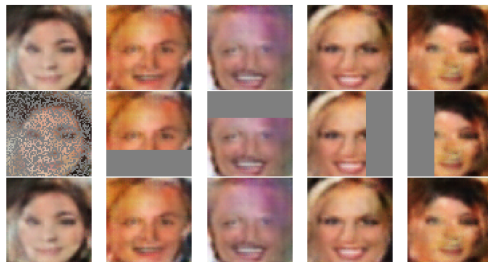

Figure 6: recovery of latent codes on a generative network trained on the CelebA dataset. From top to bottom: ground truth output images; partial measurements with different sampling masks; reconstructed image using the recovered latent codes from partial measurements. The recovery of latent codes in these examples is perfect, using simple gradient descent.

**CelebFaces:** A similar study is conducted on a generative network trained on the CelebFaces [12] dataset. We rescale the raw grayscale images from the MNIST dataset [11] to size of $64 \times 64$. A similar network architecture to previous MNIST experiment is adopted, but both the generative model and the discriminator have 4 layers rather than 3. The images are showcased in Figure 6.

Note that the probability of exact recovery increases with the number of measurements. The minimum number of measurements required for exact recovery, however, depends on the network architecture, the weights, and the sampling spatial patterns. The mathematical characterization for minimal number of measurements remains a challenging open question.

## 5   Conclusion

In this work we prove rigorously that a 2-layer ReLU convolutional generative neural network is invertible, even when only partial output is observed. This result provides a sufficient condition for the generator network to be one-to-one, which avoids the mode collapse problem in training of GAN. We empirically demonstrate that the same conclusion holds even if the generative models have other nonlinear activation functions (LeakyReLU, Sigmoid and Tanh) and multiple layers. The same proof technique can be potentially generalized to multi-layer networks. Some interesting future research directions include rigorous proofs for leaky ReLUs and other activation functions, subgaussian network weights, as well as inversion under noisy measurements.

## Footnotes

[4]Note that this network inversion problem happens at the inference stage, and thus is independent of the training process.

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
