[Supplementary Material]

# Supplementary Materials
# Invertibility of Convolutional Generative Networks from Partial Measurements

**Fangchang Ma***
MIT
fcma@mit.edu

**Ulas Ayaz***
MIT
uayaz@mit.edu
uayaz@lyft.com

**Sertac Karaman**
MIT
sertac@mit.edu

This document contains proof of the theorem and lemmas in the main document [2]. Cross references to sections, figures, equations, theorems and lemmas in [2] use the original numbering as they appear there; whereas references to this supplementary document start with *SM-* prefix.

## Sup.1   Notation

Table Sup.1: Summary of notation

| Notation | Interpretation |
|---|---|
| $\mathbf{x}, \mathbf{z}$ | two arbitrary, non-zero vectors |
| $\mathbf{x}^{(i)}, \mathbf{z}^{(i)}$ | the corresponding output vectors at layer $i \in \{0, 1, 2\}$ |
| $x_j^{(i)}, z_j^{(i)}$ | the $j^{\text{th}}$ block of $\mathbf{x}^{(i)}, \mathbf{z}^{(i)}$, no permutation |
| $\tilde{\mathbf{x}}^{(i)}, \tilde{\mathbf{z}}^{(i)}$ | the permuted vectors at layer $i \in \{0, 1, 2\}$ |
| $\tilde{x}_j^{(i)}, \tilde{z}_j^{(i)}$ | the $j^{\text{th}}$ block of $\mathbf{x}^{(i)}, \mathbf{z}^{(i)}$ after permutation |
| $D_0, D_1, D_2$ | number of blocks in a vector (before $1^{\text{st}}$ layer, after permutation, and after $2^{\text{nd}}$ layer, respectively) |
| $\mathbf{W}_1, \mathbf{W}_2$ | block weight matrices at each convolution layer |
| $\mathbf{W}_{i,+,\mathbf{z}}$ | effective weight matrices with ReLU taken into account, when input vector is $\mathbf{z}$ |
| $\mathbf{Q}_{\mathbf{x},\mathbf{z}}$ | expectation of $\mathbf{W}_{+,\mathbf{x}}^{\mathsf{T}} \mathbf{W}_{+,\mathbf{z}}$ |
| $\boldsymbol{\theta}^{(i)}$ | the angle between two different vectors $\mathbf{x}, \mathbf{z}$ at layer $i$ |
| $\theta_j^{(i)}$ | the angle between the $j^{\text{th}}$ blocks of $x_j^{(i)}, z_j^{(i)}$ at layer $i$, without permutation |
| $\tilde{\theta}_j^{(i)}$ | the angle between the $j^{\text{th}}$ blocks of $\tilde{x}_j^{(i)}, \tilde{z}_j^{(i)}$ at layer $i$, with permutation |
| $\tilde{h}_{\mathbf{x},\mathbf{z}}$ | a random vector that $\mathbf{W}_{1,+,\mathbf{x}}^{\mathsf{T}} \mathbf{W}_{2,+,\mathbf{x}}^{\mathsf{T}} \mathbf{W}_{2,+,\mathbf{z}} \mathbf{W}_{1,+,\mathbf{z}} \mathbf{z}$ concentrates around |
| $h_{\mathbf{z},\mathbf{z}^\diamond}$ | the perturbation of $\tilde{h}_{\mathbf{z},\mathbf{z}^\diamond}$ around its mean $\mathbf{z}/4$ |
| $S_{\epsilon,\mathbf{z}^\diamond}$ | a small region outside which the perturbation $h_{\mathbf{z},\mathbf{z}^\diamond}$ is very small |
| $v_{\mathbf{z},\mathbf{z}^\diamond}$ | descent direction at $\mathbf{z}$ |

Before continuing to the proof of Theorem 1, we introduce some notation.

Let $\mathrm{I}_n$ be an $n \times n$ identity matrix. If dimension is not specified, we assume it is clear from the context. Let $\mathrm{diag}(Az > 0)$ be a diagonal matrix, where $(i,i)^{\text{th}} = 1$ if $(Az)_i > 0$, and 0 otherwise. Let $\mathcal{B}(z, r)$ be an Euclidean ball of radius $r$ centered at $z$. Let $W_{1,+,z} = \mathrm{diag}(W_1 z > 0)W_1$ and $W_{2,+,z} = \mathrm{diag}(W_2 W_{1,+,z} z > 0)W_2$. For matrices, $\|A\|$ denotes the spectral norm. Let $S^{k-1}$ be the unit sphere in $\mathbb{R}^k$. A block vector $\mathbf{z} = [z_i]_i^n \in \mathbb{R}^{kn}$ is a concatenation of $n$ vectors, each of

size $k$, and uses boldface notation. Similarly, a diagonal block matrix is denoted $\mathbf{W} = [W_i]_i^n$, with matrices $\{W_i\}$ on it diagonal. For any nonzero $z \in \mathbb{R}^k$, let $h_x = \frac{x}{\|x\|_2}$. For block vector $\mathbf{z} = [z_i]_i^n$, let $\bar{\mathbf{z}} = [\bar{z}_i]_{i=1}^n$. For fixed $x, z \in \mathbb{R}^k$, let $M_{h_x \leftrightarrow \bar{z}}$ be the matrix such that $M_{h_x \leftrightarrow \bar{z}} h_x = \bar{z}$, $M_{h_x \leftrightarrow \bar{z}} \bar{z} = h_x$ and $M_{h_x \leftrightarrow \bar{z}} v = 0$ for all $v \in \text{span}(\{x, z\})^\perp$. Then given block vectors $\mathbf{x} = [x_i]_{i=1}^n$, $\mathbf{z} = [z_i]_{i=1}^n$, let $\mathbf{M}_{\bar{\mathbf{x}} \leftrightarrow \bar{\mathbf{z}}} = [M_{h_{x_i} \leftrightarrow h_{z_i}}]_{i=1}^n$. Denote the block identity matrix $\mathbf{I} = [\mathbf{I}_k]_{i=1}^n$. Let $\angle(x, z)$ be the angle between two vectors $x$ and $z$.

Recall that the weight matrix $\mathbf{W}_i$ each for layers $i = 1, 2$ is assumed to be permuted to be a block matrix, as illustrated in Figure Sup.1(b). A corresponding permutation is also applied to the input vectors of each layers.

Specifically, assume an input block vector $\mathbf{z} = \mathbf{z}^{(0)} = [z_j^{(0)}]_{j=1}^{D_0}$, then the output of the first transposed convolution layer is $\mathbf{z}^{(1)} = \mathbf{W}_{1,+,\mathbf{z}} \mathbf{z}^{(0)}$ which also has $D_0$ channels (blocks). Before the second convolution layer, we apply another permutation such that the new vector $\tilde{\mathbf{z}}^{(1)} = \text{Perm}(\mathbf{z}^{(1)})$ now has $D_1$ blocks. $\tilde{\mathbf{z}}^{(1)}$ is then fed as an input to the second transpose convolution layer, resulting in an output $\mathbf{z}^{(2)} = \mathbf{W}_{2,+,\mathbf{z}} \tilde{\mathbf{z}}^{(1)}$.

In addition, let $\boldsymbol{\theta}^{(i)} = \angle(\mathbf{x}^{(i)}, \mathbf{z}^{(i)})$ denote the angle between two different vectors $\mathbf{x}^{(i)}, \mathbf{z}^{(i)}$ at the $i^{\text{th}}$ layer. In particular, let $\theta_j^{(i)} = \angle(x_j^{(i)}, z_j^{(i)})$ denote the angle between the $j^{\text{th}}$ blocks of the two vectors. If the vectors are permuted, we use $\tilde{\theta}_j^{(i)} = \angle(\tilde{x}_j^{(i)}, \tilde{z}_j^{(i)})$. We also introduce the notation $\odot$ for multiplication of a regular vector $a \in \mathbb{R}^n$ and a block vector $\mathbf{z} = [z_j]_{j=1}^n$ in the following way:

$$a \odot \mathbf{z} = [a_j z_j]_{j=1}^n.$$

We use big-$\mathcal{O}(\cdot)$ notation to denote the order of magnitude for a variable. Finally, we also use $\lesssim, \gtrsim$ and $\simeq$ when the inequalities and equalities are up to a small universal constant $\epsilon$ which may not be specified. For instance $x \simeq y$ indicates that $x = y + \mathcal{O}(\epsilon)$. All vector and angle notations used are summarized in Table Sup.1.

## Sup.2 Proof for Theorem 1

Figure Sup.1: Illustration of a single transposed convolution operation. $k_i^j$ stands for $i^{th}$ filter kernel for the $j^{th}$ input channel. $z$ and $x$ denote the input and output signals, respectively. (a) The standard transposed convolution represented as linear multiplication. (b) With proper row and column permutations, the permuted weight matrix has a repeating block structure.

Proof mostly follows the arguments in the recent paper by [1]. As we discussed in Section 2, the weight matrix $W_1 \in \mathbb{R}^{C_1 D_1 \times C_0 D_0}$ of the first layer of the network can be arranged as a block matrix $\mathbf{W}_1 = [W]_{i=1}^{D_0}$ where $W \in \mathbb{R}^{C_1 \ell \times C_0}$ is a Gaussian matrix repeating in each block, see Figure Sup.1. In the rest of the proof we will use this arrangement of the matrix. Note that this effectively means a

permuation of the vectors after each layer. This has to be handled carefully througout the proof. By assumption, the sampling matrix $A$ is an identity matrix, so the cost function can be written as

$$J(\mathbf{z}) = \frac{1}{2}\|G(\mathbf{z}^\diamond) - G(\mathbf{z})\|_2^2.$$

The operation of the first layer on an input signal $\mathbf{z} = [z_i]_{i=1}^{D_0}$ is

$$\mathbf{W}_{1,+,\mathbf{z}}\mathbf{z} = \sigma(\mathbf{W}_1 \mathbf{z}).$$

In general operation of the generator network can be written as

$$G(\mathbf{z}) = \mathbf{W}_{2,+,\mathbf{z}}\mathbf{W}_{1,+,\mathbf{z}}\mathbf{z}.$$

**Remark Sup.0.** *The matrix* $\mathbf{W}_{1,+,\mathbf{z}}$ *captures the operation of ReLU activation combined with weights of each layer, hence it will be instrumental in the rest of the proof. We note that that in the case of Leaky ReLU activation* $L(x) = \begin{cases} x & \text{if } x \geqslant 0 \\ \alpha x & \text{if } x < 0 \end{cases}$, *the output of each layer is*

$$L(\mathbf{W}_1\mathbf{z}) = \operatorname{diag}(\mathbf{W}_1\mathbf{z} > 0)\mathbf{W}_1\mathbf{z} + \alpha \operatorname{diag}(\mathbf{W}_1\mathbf{z} < 0)\mathbf{W}_1\mathbf{z} = \alpha\mathbf{W}_1\mathbf{z} + (1-\alpha)\mathbf{W}_{1,+,\mathbf{z}}\mathbf{z}.$$

In the rest of the proof we assume the input vectors are in the block form as well as the weight matrices and denote them in boldface. This does not change the operation of the neural network. Next we prove a central technical lemma which concerns concentration of the matrix $\mathbf{W}_{1,+,\mathbf{x}}^{\mathsf{T}}\mathbf{W}_{1,+,\mathbf{z}}$. First we define the following matrices. For any nonzero $x, z \in \mathbb{R}^n$ with angle $\theta_{x,z} = \angle(x, z)$ between , let

$$Q_{x,z} := \frac{\pi - \theta_{x,z}}{2\pi}\mathrm{I}_n + \frac{\sin\theta_{x,z}}{2\pi}M_{h_x \leftrightarrow \bar{z}}. \tag{Sup.1}$$

Similarly for two block vectors $\mathbf{x} = [x_i]_{i=1}^n$ and $\mathbf{z} = [z_i]_{i=1}^n$, define the block matrix

$$\mathbf{Q}_{\mathbf{x},\mathbf{z}} := [Q_{x_i,z_i}]_{i=1}^n. \tag{Sup.2}$$

The following result appears in [1].

**Lemma Sup.1** (Lemma 5 [1]). *Fix* $\epsilon \in (0, 1)$. *Let* $A \in \mathbb{R}^{n \times k}$ *have i.i.d.* $\mathcal{N}(0, 1/n)$ *entries. If* $n > ck \log k$, *then with probability at least* $1 - 8ne^{-\gamma k}$,

$$\forall x, z \in \mathbb{R}^k, \|A_{+,x}^{\mathsf{T}}A_{+,z} - Q_{x,z}\| \leqslant \epsilon. \tag{Sup.3}$$

*When* $x = y$, *it holds*

$$\forall x \neq 0, \|A_{+,x}^{\mathsf{T}}A_{+,x} - \mathrm{I}_n/2\| \leqslant \epsilon. \tag{Sup.4}$$

*Here* $c, \gamma$ *depends only on* $\epsilon$.

Here the matrix $Q_{x,z}$ happens to be the expectation of the matrix $A_{+,x}^{\mathsf{T}}A_{+,z}$. This can be shown by an elementary calculation. We can now state the central technical lemma.

**Lemma Sup.2.** *Fix* $\epsilon \in (0, 1)$. *Let* $\mathbf{W} = [W]_{i=1}^{D_0}$ *where* $W \in \mathbb{R}^{C_1\ell \times C_0}$ *have i.i.d.* $\mathcal{N}(0, 1/C_1\ell)$. *If* $C_1\ell \gtrsim_\epsilon C_0 \log C_0$, *then with probability at least* $1 - 8D_0\ell C_1\, e^{-\gamma C_0}$,

$$\forall \mathbf{x}, \mathbf{z} \in \mathbb{R}^{C_0 D_0}, \|\mathbf{W}_{+,\mathbf{x}}^{\mathsf{T}}\mathbf{W}_{+,\mathbf{z}} - \mathbf{Q}_{\mathbf{x},\mathbf{z}}\| \leqslant \epsilon. \tag{Sup.5}$$

*When* $x = y$, *it holds*

$$\forall \mathbf{x} \neq 0, \|\mathbf{W}_{+,\mathbf{x}}^{\mathsf{T}}\mathbf{W}_{+,\mathbf{x}} - \mathbf{I}/2\| \leqslant \epsilon. \tag{Sup.6}$$

*Here* $c, \gamma$ *depends only on* $\epsilon$.

Lemma Sup.2 is crucial to the rest of the proof. We note that the conditions of Theorem 1 are almost identical to the ones of Lemma Sup.2. In other words, the concentration of weight matrices as given in (4) and (5) are enough to imply the existence of a strict descent direction for the cost function $J(x)$ outside of two small neighborhoods. We now prove the lemma.

*Proof.* Observe that for block vectors $\mathbf{x} = [x_i]_{i=1}^{D_0}$ and $\mathbf{z} = [z_i]_{i=1}^{D_0}$ we can write

$$\mathbf{W}_{1,+,\mathbf{x}}^{\mathsf{T}}\mathbf{W}_{1,+,\mathbf{z}} = [W_{+,x_i}^{\mathsf{T}}W_{+,z_i}]_{i=1}^{D_0}. \tag{Sup.7}$$

We know from spectral norm of block matrices that

$$\|\mathbf{W}_{1,+,\mathbf{x}}^{\mathsf{T}}\mathbf{W}_{1,+,\mathbf{z}} - \mathbf{Q}_{\mathbf{x},\mathbf{z}}\| = \max_{i=1,\dots,D_0} \|W_{+,x_i}^{\mathsf{T}}W_{+,z_i} - Q_{x_i,z_i}\|. \tag{Sup.8}$$

Lemma Sup.1 implies that if $C_1\ell \gtrsim C_0 \log C_0$ then with probability at least $1 - 8C_1\ell e^{-\gamma C_0}$, it holds that $\|W_{+,x_i}^{\mathsf{T}}W_{+,z_i} - Q_{x_i,z_i}\| \leqslant \epsilon$. A union bound argument yields

$$\mathbb{P}(\|\mathbf{W}_{1,+,\mathbf{x}}^{\mathsf{T}}\mathbf{W}_{1,+,\mathbf{z}} - \mathbf{Q}_{\mathbf{x},\mathbf{z}}\| \leqslant \epsilon) \leqslant \sum_{i=1}^{D_0} \mathbb{P}(\|W_{+,x_i}^{\mathsf{T}}W_{+,z_i} - Q_{x_i,z_i}\| \leqslant \epsilon) \leqslant 8D_0C_1\ell\, e^{-\gamma C_0}.$$

$\square$

Next, we present another useful result that controls how $\mathbf{W}_{1,+,\mathbf{x}}$ distorts the angle between two vectors $\mathbf{x}, \mathbf{z}$. Let:

$$g(\theta) := \cos^{-1}\left(\frac{(\pi - \theta)\cos\theta + \sin\theta}{\pi}\right).$$

First we borrow another result from [1].

**Lemma Sup.3** (Lemma 23 [1])**.** *Fix $\epsilon \in (0, 0.1)$. Let the conditions of Lemma Sup.1 hold and A satisfy* (Sup.3)*. For $x, z$ denote $\theta_0 = \angle(x, z)$ and $\theta_1 := \angle(A_{+,x}x, A_{+,z}z)$. Then*

$$|\theta_1 - g(\theta_0)| \leqslant 4\sqrt{\epsilon}.$$

This lemma shows that a Gaussian matrix combined with ReLU operation preserves the angle between vectors up to function $g(\cdot)$. Our next lemma uses this result.

**Lemma Sup.4.** *Fix $\epsilon < 1/(16\pi)^2$. Assume that weight matrices $\mathbf{W}_1$ and $\mathbf{W}_2$ satisfy* (Sup.5) *with constant $\epsilon$. Then it holds for all $\mathbf{x}, \mathbf{z} \neq 0$ that*

$$\langle \mathbf{W}_{2,+,\mathbf{x}}\mathbf{W}_{1,+,\mathbf{x}}\mathbf{x}, \mathbf{W}_{2,+,\mathbf{z}}\mathbf{W}_{1,+,\mathbf{z}}\mathbf{z}\rangle > 0.$$

*Proof.* We operate under the assumptions that the weight matrices $W_j$ for layer $i = 1, 2$ satisfy Equation (Sup.4) and Equation (Sup.6). In particular these equations imply that for layer $i = 1, 2$ and all input vector $z, \mathbf{z} \neq 0$,

$$\frac{1}{2} - \epsilon \leqslant \|W_{i,+,z}\|^2 \leqslant \frac{1}{2} + \epsilon \tag{Sup.9}$$

$$\frac{1}{2} - \epsilon \leqslant \|\mathbf{W}_{i,+,\mathbf{z}}\|^2 \leqslant \frac{1}{2} + \epsilon. \tag{Sup.10}$$

Since $\mathbf{z}^{(1)} = \mathbf{W}_{1,+,\mathbf{z}}\mathbf{z}^{(0)}$, it follows that for all blocks $j = 1, \dots, D_0$ that

$$\sqrt{\frac{1}{2} - \epsilon}\, \|z_j^{(0)}\|_2 \leqslant \|z_j^{(1)}\|_2 \leqslant \sqrt{\frac{1}{2} + \epsilon}\, \|z_j^{(0)}\|_2 \tag{Sup.11}$$

$$\sqrt{\frac{1}{2} - \epsilon}\, \|\mathbf{z}^{(0)}\|_2 \leqslant \|\mathbf{z}^{(1)}\|_2 \leqslant \sqrt{\frac{1}{2} + \epsilon}\, \|\mathbf{z}^{(0)}\|_2.$$

The same statements hold true for $\mathbf{x}$ and $x_j^{(i)}$ as well. Consequently, by dividing the inequalities we have for $\mathbf{x}$ and $\mathbf{z}$ that

$$\sqrt{\frac{1 - 2\epsilon}{1 + 2\epsilon}}\, \frac{\|x_j^{(0)}\|_2}{\|z_j^{(0)}\|_2} \leqslant \frac{\|x_j^{(1)}\|_2}{\|z_j^{(1)}\|_2} \leqslant \sqrt{\frac{1 + 2\epsilon}{1 - 2\epsilon}}\, \frac{\|x_j^{(0)}\|_2}{\|z_j^{(0)}\|_2} \tag{Sup.12}$$

$$\sqrt{\frac{1 - 2\epsilon}{1 + 2\epsilon}}\, \frac{\|\mathbf{x}^{(0)}\|_2}{\|\mathbf{z}^{(0)}\|_2} \leqslant \frac{\|\mathbf{x}^{(1)}\|_2}{\|\mathbf{z}^{(1)}\|_2} \leqslant \sqrt{\frac{1 + 2\epsilon}{1 - 2\epsilon}}\, \frac{\|\mathbf{x}^{(0)}\|_2}{\|\mathbf{z}^{(0)}\|_2}. \tag{Sup.13}$$

Figure Sup.2: Plot of function $g(\cdot)$ defined in Section Sup.2

We assume without loss of generality that the input vectors $\mathbf{x}^{(0)} = [x_j^{(0)}]_{j=1}^{D_0}$ and $\mathbf{z}^{(0)} = [z_j^{(0)}]_{j=1}^{D_0}$ are block normalized, i.e., $\|x_j^{(0)}\|_2 = \|z_j^{(0)}\|_2 = 1$. We have

$$\langle \mathbf{x}^{(1)}, \mathbf{z}^{(1)} \rangle = \sum_j \langle x_j^{(1)}, z_j^{(1)} \rangle =$$

$$\|\mathbf{x}^{(1)}\|_2 \|\mathbf{z}^{(1)}\|_2 \cos \boldsymbol{\theta}^{(1)} = \sum_j \|x_j^{(1)}\|_2 \|z_j^{(1)}\|_2 \cos \theta_j^{(1)}$$

$$\geqslant \min_j \cos \theta_j^{(1)} \sum_j \|x_j^{(1)}\|_2 \|z_j^{(1)}\|_2. \tag{Sup.14}$$

Using the fact that input vectors are block normalized, it follows from the first inequality in (Sup.12) that

$$\sqrt{\frac{1-2\epsilon}{1+2\epsilon}} \|z_j^{(1)}\|_2^2 \leqslant \|x_j^{(1)}\|_2 \|z_j^{(1)}\|_2 \tag{Sup.15}$$

and from the second inequality in (Sup.13) that

$$\|\mathbf{x}^{(1)}\|_2 \|\mathbf{z}^{(1)}\|_2 \leqslant \sqrt{\frac{1+2\epsilon}{1-2\epsilon}} \|\mathbf{z}^{(1)}\|_2^2 \tag{Sup.16}$$

Combining (Sup.14), (Sup.15) and (Sup.16) yields that

$$\sqrt{\frac{1+2\epsilon}{1-2\epsilon}} \|\mathbf{z}^{(1)}\|_2^2 \cos \boldsymbol{\theta}^{(1)} \geqslant$$

$$\sqrt{\frac{1-2\epsilon}{1+2\epsilon}} \min_j \cos \theta_j^{(1)} \sum_j \|z_j^{(1)}\|_2^2$$

which in turn implies that

$$(1 + 8\epsilon) \cos \boldsymbol{\theta}^{(1)} \geqslant \min_j \cos \theta_j^{(1)} = \cos \theta_{\hat{i}}^{(1)}, \tag{Sup.17}$$

for some $\hat{i}$. Here we used the fact that $\frac{1+v}{1-v} \leqslant 1 + 4v$ for $0 \leqslant z \leqslant \frac{1}{2}$. Lemma Sup.3 implies that $|\theta_{\hat{i}}^{(1)} - g(\theta_{\hat{i}}^{(0)})| \leqslant 4\sqrt{\epsilon}$. Since $g(\theta_{\hat{i}}^{(0)}) < \frac{\pi}{4}$ (see Figure Sup.2), we have $0 \leqslant \theta_{\hat{i}}^{(1)} \leqslant \frac{\pi}{4} + 4\sqrt{\epsilon}$. Then for small enough $\epsilon$, (Sup.17) implies that

$$\cos \boldsymbol{\theta}^{(1)} \geqslant \frac{\cos \theta_{\hat{i}}^{(1)}}{1 + 8\epsilon} \geqslant \frac{\cos(\frac{\pi}{4} + 4\sqrt{\epsilon})}{1 + 8\epsilon} > 0.6.$$

The last inequality comes from the fact that $\frac{\cos(\frac{\pi}{4}+4\sqrt{\epsilon})}{1+8\epsilon}$ is monotonically decreasing with small $\epsilon$, so the minimum can be computed and is roughly equals to $0.646$, since $\epsilon < 1/(16\pi)^2$. The exact number doesn't affect the final results, because we only want to bound $\cos \boldsymbol{\theta}^{(1)}$ away from $0$.

Now we proceed to the second layer of the network. As explained in the Section Sup.1, we first reorder vectors $\tilde{\mathbf{x}}^{(1)} = [\tilde{x}_j^{(1)}]_{j=1}^{D_1}$ and $\tilde{\mathbf{z}}^{(1)} = [\tilde{z}_j^{(1)}]_{j=1}^{D_1}$. We expand the dot product of $\mathbf{x}^{(1)}$ and $\mathbf{z}^{(1)}$ similarly as before

$$\|\mathbf{x}^{(1)}\|_2 \|\mathbf{z}^{(1)}\|_2 \cos \boldsymbol{\theta}^{(1)} = \sum_j \|\tilde{x}_j^{(1)}\|_2 \|\tilde{z}_j^{(1)}\|_2 \cos \tilde{\theta}_j^{(1)}.$$

Define the set of indices
$$I = \{j : \|\tilde{x}_j^{(1)}\|_2 \neq 0 \text{ and } \|\tilde{x}_j^{(1)}\|_2 \neq 0\}.$$

Then we can continue

$$\|\mathbf{x}^{(1)}\|_2 \|\mathbf{z}^{(1)}\|_2 \cos \boldsymbol{\theta}^{(1)} = \sum_{j \in I} \|\tilde{x}_j^{(1)}\|_2 \|\tilde{z}_j^{(1)}\|_2 \cos \tilde{\theta}_j^{(1)}$$
$$\leqslant \max_{j \in I} \cos \tilde{\theta}_j^{(1)} \sum_{j \in I} \|\tilde{x}_j^{(1)}\|_2 \|\tilde{z}_j^{(1)}\|_2$$
$$\leqslant \max_{j \in I} \cos \tilde{\theta}_j^{(1)} \|\tilde{\mathbf{x}}^{(1)}\|_2 \|\tilde{\mathbf{z}}^{(1)}\|_2$$

where we used Cauchy-Schwartz inequality in the last line. Since the reordering does not change the norm of the block vectors, we have $\|\mathbf{x}^{(1)}\|_2 \|\mathbf{z}^{(1)}\|_2 = \|\tilde{\mathbf{x}}^{(1)}\|_2 \|\tilde{\mathbf{z}}^{(1)}\|_2$. Consequently, it follows that $\cos \boldsymbol{\theta}^{(1)} \leqslant \max_{j \in I} \cos \tilde{\theta}_j^{(1)}$. In other words, for some $\hat{j} \in I$,

$$0.6 < \cos \boldsymbol{\theta}^{(1)} \leqslant \cos \tilde{\theta}_{\hat{j}}^{(1)}. \tag{Sup.18}$$

Recall that after the second transposed convolution layer of the network, we have $z_j^{(2)} = W_2 \tilde{z}_j^{(1)}$. Since similar relation as in (Sup.11) holds for second layer as well and we have $\|\tilde{x}_{\hat{j}}^{(1)}\|_2 > 0$ and $\|\tilde{z}_{\hat{j}}^{(1)}\|_2 > 0$ as $\hat{j} \in I$, it follows that

$$\|\tilde{x}_{\hat{j}}^{(2)}\|_2 > 0 \text{ and } \|\tilde{z}_{\hat{j}}^{(2)}\|_2 > 0. \tag{Sup.19}$$

By invoking Lemma Sup.3 once again, we have $|\theta_{\hat{j}}^{(2)} - g(\tilde{\theta}_{\hat{j}}^{(1)})| \leqslant 4\sqrt{\epsilon}$. Combining this with (Sup.18) yields

$$\cos \theta_{\hat{j}}^{(2)} \geqslant 0.6. \tag{Sup.20}$$

Finally we arrive at the desired result

$$\langle \mathbf{W}_{2,+,\mathbf{x}} \mathbf{W}_{1,+,\mathbf{x}} \mathbf{x}^{(0)}, \mathbf{W}_{2,+,\mathbf{z}} \mathbf{W}_{1,+,\mathbf{z}} \mathbf{z}^{(0)} \rangle$$
$$= \langle \mathbf{x}^{(2)}, \mathbf{z}^{(2)} \rangle = \sum_j \|x_j^{(2)}\|_2 \|z_j^{(2)}\|_2 \cos \theta_j^{(2)}$$
$$\geqslant \|x_{\hat{j}}^{(2)}\|_2 \|z_{\hat{j}}^{(2)}\|_2 \cos \theta_{\hat{j}}^{(2)} > 0$$

which follows from (Sup.19) and (Sup.20).

$\square$

### Sup.2.1    Additional Lemmas

Recall that the block weight matrix is defined as $\mathbf{W}_1 = [W_1]_{j=1}^{D_0}$ where $W_1 \in \mathbb{R}^{C_1 \ell \times C_0}$. Similarly, $\mathbf{W}_2 = [W_2]_{j=1}^{D_1}$ where $W_2 \in \mathbb{R}^{C_2 \ell \times C_1}$. In the following lemma, we extend the concentration of matrix products in Lemma Sup.2 to 2-layer networks.

**Lemma Sup.5.** *Fix $\epsilon \in (0,1)$. Let $W_1$ have i.i.d. $\mathcal{N}(0, 1/C_1\ell)$ weights and $W_2$ have i.i.d. $\mathcal{N}(0, 1/C_2\ell)$ weights. Assume the conditions of Lemma Sup.2 hold, then with high probability for all $\mathbf{z} \neq \mathbf{0}$,*

$$\|\mathbf{W}_{1,+,\mathbf{z}}^\mathsf{T}\mathbf{W}_{2,+,\mathbf{z}}^\mathsf{T}\mathbf{W}_{2,+,\mathbf{z}}\mathbf{W}_{1,+,\mathbf{z}} - \mathbf{I}/4\| \leqslant 2\epsilon. \tag{Sup.21}$$

*Proof.* From Lemma Sup.2, we know that for all $\mathbf{z} \neq \mathbf{0}$,

$$\|\mathbf{W}_{1,+,\mathbf{z}}^\mathsf{T}\mathbf{W}_{1,+,\mathbf{z}} - \mathbf{I}/2\| \leqslant \epsilon$$
$$\|\mathbf{W}_{2,+,\mathbf{z}}^\mathsf{T}\mathbf{W}_{2,+,\mathbf{z}} - \mathbf{I}/2\| \leqslant \epsilon$$

Also, from Equation (Sup.10) we have

$$\|\mathbf{W}_{1,+,\mathbf{z}}\|^2 \leqslant \frac{1}{2} + \epsilon.$$

Using these two along with the triangle inequality, it follows that

$$\begin{aligned}
&\|\mathbf{W}_{1,+,\mathbf{z}}^\mathsf{T}\mathbf{W}_{2,+,\mathbf{z}}^\mathsf{T}\mathbf{W}_{2,+,\mathbf{z}}\mathbf{W}_{1,+,\mathbf{z}} - \mathbf{I}/4\| \\
&= \left\| \left( \mathbf{W}_{1,+,\mathbf{z}}^\mathsf{T}\mathbf{W}_{2,+,\mathbf{z}}^\mathsf{T}\mathbf{W}_{2,+,\mathbf{z}}\mathbf{W}_{1,+,\mathbf{z}} - \frac{1}{2}\mathbf{W}_{1,+,\mathbf{z}}^\mathsf{T}\mathbf{W}_{1,+,\mathbf{z}} \right) \right. \\
&\qquad \left. + \left( \frac{1}{2}\mathbf{W}_{1,+,\mathbf{z}}^\mathsf{T}\mathbf{W}_{1,+,\mathbf{z}} - \mathbf{I}/4 \right) \right\| \\
&\leqslant \|\mathbf{W}_{1,+,\mathbf{z}}^\mathsf{T}(\mathbf{W}_{2,+,\mathbf{z}}^\mathsf{T}\mathbf{W}_{2,+,\mathbf{z}} - \mathbf{I}/2)\mathbf{W}_{1,+,\mathbf{z}}\| \\
&\qquad + \frac{1}{2}\|\mathbf{W}_{1,+,\mathbf{z}}^\mathsf{T}\mathbf{W}_{1,+,\mathbf{z}} - \mathbf{I}/2\| \\
&\leqslant \|\mathbf{W}_{1,+,\mathbf{z}}\|^2\|\mathbf{W}_{2,+,\mathbf{z}}^\mathsf{T}\mathbf{W}_{2,+,\mathbf{z}} - \mathbf{I}/2\| \\
&\qquad + \frac{1}{2}\|\mathbf{W}_{1,+,\mathbf{z}}^\mathsf{T}\mathbf{W}_{1,+,\mathbf{z}} - \mathbf{I}/2\| \\
&\leqslant \left( \frac{1}{2} + \epsilon \right)\epsilon + \frac{1}{2}\epsilon \leqslant 2\epsilon.
\end{aligned}$$

$\square$

Given block vectors $\mathbf{x}$ and $\mathbf{z}$, we define the following vector

$$\tilde{h}_{\mathbf{x},\mathbf{z}} := \left[ \frac{\pi - \tilde{\theta}_j^{(1)}}{2\pi} \right]_{j=1}^{D_1} \odot \left( \left[ \frac{\pi - \theta_j^{(0)}}{2\pi} \right]_{j=1}^{D_0} \odot \mathbf{z} + \left[ \frac{\sin\theta_j^{(0)}\|z_j^{(0)}\|_2}{2\pi\|x_j^{(0)}\|_2} \right]_{j=1}^{D_0} \odot \mathbf{x} \right) + \left[ \frac{\sin\tilde{\theta}_j^{(1)}\|\tilde{z}_j^{(1)}\|_2}{2\pi\|\tilde{x}_j^{(1)}\|_2} \right]_{j=1}^{D_1} \odot \mathbf{x} \tag{Sup.22}$$

where $\tilde{\theta}_j^{(1)}, \tilde{x}_j^{(1)}, \tilde{z}_j^{(1)}$ are defined as in Section Sup.1 and Table Sup.1. Next we show that $\mathbf{W}_{1,+,\mathbf{x}}^\mathsf{T}\mathbf{W}_{2,+,\mathbf{x}}^\mathsf{T}\mathbf{W}_{2,+,\mathbf{z}}\mathbf{W}_{1,+,\mathbf{z}}\mathbf{z}$ concentrates around this random vector $\tilde{h}_{\mathbf{x},\mathbf{z}}$.

**Lemma Sup.6.** *Assume $\mathbf{W}_1$ and $\mathbf{W}_2$ satisfy the conditions of Lemma Sup.2. The for all $\mathbf{x} \neq 0, \mathbf{z} \neq 0$ we have*

$$\|\mathbf{W}_{1,+,\mathbf{x}}^\mathsf{T}\mathbf{W}_{2,+,\mathbf{x}}^\mathsf{T}\mathbf{W}_{2,+,\mathbf{z}}\mathbf{W}_{1,+,\mathbf{z}}\mathbf{z} - \tilde{h}_{\mathbf{x},\mathbf{z}}\|_2 \lesssim \epsilon\max\{\|\mathbf{x}\|_2, \|\mathbf{z}\|_2\}.$$

*Proof.* We expand

$$\begin{aligned}
\mathbf{W}_{1,+,\mathbf{x}}^\mathsf{T}\mathbf{W}_{2,+,\mathbf{x}}^\mathsf{T}&\mathbf{W}_{2,+,\mathbf{z}}\mathbf{W}_{1,+,\mathbf{z}}\mathbf{z} = \\
&\underbrace{\mathbf{W}_{1,+,\mathbf{x}}^\mathsf{T}[\mathbf{W}_{2,+,\mathbf{x}}^\mathsf{T}\mathbf{W}_{2,+,\mathbf{z}} - \mathbf{Q}_{\mathbf{x}^{(1)},\mathbf{z}^{(1)}}]\mathbf{W}_{1,+,\mathbf{z}}\mathbf{z}}_{T_1} \\
&+ \underbrace{\left[ \frac{\pi - \tilde{\theta}_j^{(1)}}{2\pi} \right]_{j=1}^{D_1} \odot \mathbf{W}_{1,+,\mathbf{x}}^\mathsf{T}\mathbf{W}_{1,+,\mathbf{z}}\mathbf{z}}_{T_2} \\
&+ \underbrace{\left[ \frac{\sin\tilde{\theta}_j^{(1)}\|\tilde{z}_j^{(1)}\|_2}{2\pi\|\tilde{x}_j^{(1)}\|_2} \right]_{j=1}^{D_1} \odot \mathbf{W}_{1,+,\mathbf{x}}^\mathsf{T}\mathbf{W}_{1,+,\mathbf{x}}\mathbf{x}}_{T_3}.
\end{aligned}$$

We have

$$|T_1| \leqslant \|\mathbf{W}_{1,+,\mathbf{x}}\| \|\mathbf{W}_{1,+,\mathbf{z}}\| \|\mathbf{W}_{2,+,\mathbf{x}}^\mathsf{T}\mathbf{W}_{2,+,\mathbf{z}} - \mathbf{Q}_{\mathbf{x}^{(1)},\mathbf{z}^{(1)}}\| \|\mathbf{z}\|_2 \leqslant \epsilon \|\mathbf{z}\|_2 \qquad \text{(Sup.23)}$$

where we used (Sup.10) and Lemma Sup.2. Expanding $T_2$ we get

$$T_2 - \left[\frac{\pi - \tilde{\theta}_j^{(1)}}{2\pi}\right]_{j=1}^{D_1} \odot \mathbf{Q}_{\mathbf{x},\mathbf{z}}\mathbf{z} = \left[\frac{\pi - \tilde{\theta}_j^{(1)}}{2\pi}\right]_{j=1}^{D_1} \odot \left[\mathbf{W}_{1,+,\mathbf{x}}^\mathsf{T}\mathbf{W}_{1,+,\mathbf{z}}\mathbf{z} - \mathbf{Q}_{\mathbf{x},\mathbf{z}}\right]$$

which can be bounded as follows

$$\left| T_2 - \left[\frac{\pi - \tilde{\theta}_j^{(1)}}{2\pi}\right]_{j=1}^{D_1} \odot \mathbf{Q}_{\mathbf{x},\mathbf{z}}\mathbf{z} \right| \leqslant \|\mathbf{W}_{1,+,\mathbf{x}}^\mathsf{T}\mathbf{W}_{1,+,\mathbf{z}}\| \|\mathbf{z}\|_2 \leqslant \epsilon \|\mathbf{z}\|_2 \qquad \text{(Sup.24)}$$

where we used (Sup.5). Using the definition (Sup.1) of $\mathbf{Q}_{\mathbf{x},\mathbf{z}}$ we observe that

$$\mathbf{Q}_{\mathbf{x},\mathbf{z}}\mathbf{z} = \left[\frac{\pi - \theta_j^{(0)}}{2\pi}\right]_{j=1}^{D_0} \odot \mathbf{z} + \left[\frac{\sin\theta_j^{(0)}\|z_j^{(0)}\|_2}{2\pi\|x_j^{(0)}\|_2}\right]_{j=1}^{D_0} \odot \mathbf{x}. \qquad \text{(Sup.25)}$$

Expanding $T_3$ we get

$$\left| T_3 - \left[\frac{\sin\tilde{\theta}_j^{(1)}\|\tilde{z}_j^{(1)}\|_2}{2\pi\|\tilde{x}_j^{(1)}\|_2}\right]_{j=1}^{D_1} \odot \mathbf{x} \right| =$$

$$\left| \left[\frac{\sin\tilde{\theta}_j^{(1)}\|\tilde{z}_j^{(1)}\|_2}{2\pi\|\tilde{x}_j^{(1)}\|_2}\right]_{j=1}^{D_1} \odot \left[\mathbf{W}_{1,+,\mathbf{x}}^\mathsf{T}\mathbf{W}_{1,+,\mathbf{x}} - \mathbf{I}\right]\mathbf{x} \right| \leqslant$$

$$\|\mathbf{W}_{1,+,\mathbf{x}}^\mathsf{T}\mathbf{W}_{1,+,\mathbf{x}} - \mathbf{I}\| \|\mathbf{x}\|_2 \leqslant \epsilon \|\mathbf{x}\|_2 \qquad \text{(Sup.26)}$$

where we used (Sup.6) in the last inequality. The result follows by combining (Sup.23), (Sup.24), (Sup.25) and (Sup.26).

$\square$

Before we proceed to finishing the proof of Theorem 1, we introduce a couple more definitions and lemmas:

$$h_{\mathbf{z},\mathbf{z}^\diamond} = \frac{\mathbf{z}}{4} - \tilde{h}_{\mathbf{z},\mathbf{z}^\diamond} \qquad \text{(Sup.27)}$$

$$S_{\epsilon,\mathbf{z}^\diamond} = \left\{\mathbf{z} : \|h_{\mathbf{z},\mathbf{z}^\diamond}\|_2 \leqslant \epsilon \max(\|\mathbf{z}\|_2, \|\mathbf{z}^\diamond\|_2)\right\} \qquad \text{(Sup.28)}$$

Note that $\tilde{h}_{\mathbf{z},\mathbf{z}^\diamond}$ has already been defined in Equation (Sup.22), and $h_{\mathbf{z},\mathbf{z}^\diamond}$ is simply the perturbation of $\tilde{h}_{\mathbf{z},\mathbf{z}^\diamond}$ around its mean. $S_{\epsilon,\mathbf{z}^\diamond}$ is the region where such perturbation is small.

**Lemma Sup.7.** *Assume the conditions of Lemma Sup.5 and Lemma Sup.6 hold. Then, with high probability for all $\mathbf{z} \neq \mathbf{0}$,*

$$\|v_{\mathbf{z},\mathbf{z}^\diamond} - h_{\mathbf{z},\mathbf{z}^\diamond}\|_2 < 2\epsilon \max(\|\mathbf{z}\|_2, \|\mathbf{z}^\diamond\|_2) \qquad \text{(Sup.29)}$$

*Proof.* Recall that the descent direction $v_{\mathbf{z},\mathbf{z}^\diamond}$ is already given in Equation (Sup.39), which we repeat as follows

$$v_{\mathbf{z},\mathbf{z}^\diamond} = \underbrace{(\mathbf{W}_{2,+,\mathbf{z}}\mathbf{W}_{1,+,\mathbf{z}})^\mathsf{T}(\mathbf{W}_{2,+,\mathbf{z}}\mathbf{W}_{1,+,\mathbf{z}})}_{T_1}\mathbf{z}$$

$$- \underbrace{(\mathbf{W}_{2,+,\mathbf{z}}\mathbf{W}_{1,+,\mathbf{z}})^\mathsf{T}(\mathbf{W}_{2,+,\mathbf{z}^\diamond}\mathbf{W}_{1,+,\mathbf{z}^\diamond})}_{T_2}\mathbf{z}^\diamond$$

$$= T_1\mathbf{z} - T_2\mathbf{z}^\diamond \qquad \text{(Sup.30)}$$

Under the conditions of Lemma Sup.5, with high probability we have that

$$\|T_1 - \mathbf{I}/4\| \leqslant 2\epsilon. \qquad \text{(Sup.31)}$$

In addition, under the conditions of Lemma Sup.6, with high probability and a change of notation we have that

$$\left\| T_2 \mathbf{z}^\diamond - \tilde{h}_{\mathbf{z}, \mathbf{z}^\diamond} \right\|_2 \leqslant \epsilon \left\| \mathbf{z}^\diamond \right\|_2. \tag{Sup.32}$$

Combining the definition of $h_{\mathbf{z}, \mathbf{z}^\diamond}$ and Line (Sup.30), it follows that

$$
\begin{aligned}
\| v_{\mathbf{z}, \mathbf{z}^\diamond} - h_{\mathbf{z}, \mathbf{z}^\diamond} \|_2 &= \left\| (T_1 \mathbf{z} - T_2 \mathbf{z}^\diamond) - \left( \frac{\mathbf{z}}{4} - \tilde{h}_{\mathbf{z}, \mathbf{z}^\diamond} \right) \right\|_2 \\
&\leqslant \left\| T_1 \mathbf{z} - \frac{1}{4} \mathbf{z} \right\|_2 + \left\| T_2 \mathbf{z}^\diamond - \tilde{h}_{\mathbf{z}, \mathbf{z}^\diamond} \right\|_2 \\
&\leqslant \| T_1 - \mathbf{I}/4 \|_2 \| \mathbf{z} \|_2 + \left\| T_2 \mathbf{z}^\diamond - \tilde{h}_{\mathbf{z}, \mathbf{z}^\diamond} \right\|_2 \\
&\leqslant 2\epsilon \| \mathbf{z} \|_2 + \epsilon \| \mathbf{z}^\diamond \|_2 \\
&< 2\epsilon \max(\| \mathbf{z} \|_2, \| \mathbf{z}^\diamond \|_2)
\end{aligned}
$$

The first inequality applies triangle inequality, the second uses the simple fact that $\| Ax \| \leqslant \| A \| \| x \|_2$, and the last inequality is a result of both Line (Sup.31) and Line (Sup.32). □

Now we introduce the last lemma needed for the main proof. This lemma establishes that the set $S_{\epsilon, \mathbf{z}}$ is a subset of the union of two small neighbors around $\mathbf{z}^\diamond$ and $-\rho \mathbf{z}^\diamond$ with radius no more than $\epsilon$.

**Lemma Sup.8.** *Suppose* $4\epsilon \leqslant 1$. *Then*

$$S_{4\epsilon, \mathbf{z}^\diamond} \subset \mathcal{B}(\mathbf{z}^\diamond, c\epsilon \| \mathbf{z}^\diamond \|_2) \cup \mathcal{B}(-\rho \mathbf{z}^\diamond, d\epsilon \| \mathbf{z}^\diamond \|_2)$$

*where* $\rho, c, d$ *are universal constants.*

*Proof.* The proof mainly follows Lemma 8 in [1]. Using the definitions (Sup.22) and (Sup.27) we can rewrite $h_{\mathbf{z}, \mathbf{z}^\diamond}$ as follows:

$$h_{\mathbf{z}, \mathbf{z}^\diamond} = -\beta \mathbf{z}^\diamond + (1/4 - \alpha) \mathbf{z}$$

where the vectors $\alpha$ and $\beta$ are

$$
\begin{aligned}
\alpha &= \left[ \frac{\pi - \tilde{\theta}_j^{(1)}}{2\pi} \right]_{j=1}^{D_1} \odot \left[ \frac{\sin \theta_j^{(0)} \| z_j^{\diamond, (0)} \|_2}{2\pi \| z_j^{(0)} \|_2} \right]_{j=1}^{D_0} + \left[ \frac{\sin \tilde{\theta}_j^{(1)} \| \tilde{z}_j^{\diamond, (1)} \|_2}{2\pi \| \tilde{z}_j^{(1)} \|_2} \right]_{j=1}^{D_1}, \\
\beta &= \left[ \frac{\pi - \tilde{\theta}_j^{(1)}}{2\pi} \right]_{j=1}^{D_1} \odot \left[ \frac{\pi - \theta_j^{(0)}}{2\pi} \right]_{j=1}^{D_0}.
\end{aligned}
$$

Without loss of generality, let $\| z_j^{\diamond, (0)} \|_2 = 1$, $\mathbf{z} \in S_{\epsilon, \mathbf{z}^\diamond}$ and $M = \max(\max_j \| z_j^{(0)} \|_2, 1)$. Then definition of $S_{\epsilon, \mathbf{z}^\diamond}$ in (Sup.28) implies that $\| h_{\mathbf{z}, \mathbf{z}^\diamond} \|_2$ is small. This, in turn, implies that

$$| - \beta_j + \cos \theta_j^{(0)} (1/4 - \alpha_j) | \leqslant \epsilon M, \tag{Sup.33}$$

$$| \sin \theta_j^{(0)} (1/4 - \alpha_j) | \leqslant \epsilon M \quad \text{for all } j. \tag{Sup.34}$$

Recall that $\theta_j^{(0)}$ is the angle between $z_j^{(0)}$ and $z_j^{\diamond, (0)}$, which are the $j$-th block of the vectors $\mathbf{z}$ and $\mathbf{z}^\diamond$ respectively. For the rest of the proof, we argue as follows. Using (Sup.33) and (Sup.34), we will show that for all $j$ either

$$\theta_j^{(0)} \leqslant 2\epsilon \quad \text{and} \quad \| z_j^{(0)} \|_2 \simeq \| z_j^{\diamond, (0)} \|_2 \tag{Sup.35}$$

or

$$| \theta_j^{(0)} - \pi | \leqslant 2\epsilon \quad \text{and} \quad \| z_j^{(0)} \|_2 \simeq \rho \| z_j^{\diamond, (0)} \|_2 \tag{Sup.36}$$

holds for some constant $\rho$. This, in turn, implies that for all $j$, $z_j^{(0)}$ is either very close to $z_j^{\diamond, (0)}$ or to the polar opposite of it up to a constant $\rho$. Indeed, observe the fact that

$$\| z_j^{(0)} - z_j^{\diamond, (0)} \|_2 \leqslant \left| \| z_j^{(0)} \|_2 - \| z_j^{\diamond, (0)} \|_2 \right| + \left( \| z_j^{\diamond, (0)} \|_2 + \left| \| z_j^{(0)} \|_2 - \| z_j^{\diamond, (0)} \|_2 \right| \right) \theta_j^{(0)}.$$

This simply says that if a vector is known to have magnitude within $\Delta r$ of some other vector with magnitude $r$ and two vectors have an angle less than $\Delta\theta$ between them, then the Euclidean distance between two vectors is no more than $\Delta r + (r + \Delta r)\Delta\theta$. Then using the assumption $\|z_j^{\diamond,(0)}\|_2 = 1$ and (Sup.35) we have

$$\|z_j^{(0)} - z_j^{\diamond,(0)}\|_2 \lesssim \epsilon.$$

Given this, we can conclude that

$$\|\mathbf{z} - \mathbf{z}^\diamond\|_2^2 = \sum_{j=1}^{D_0} \|z_j^{(0)} - z_j^{\diamond,(0)}\|_2^2 \lesssim \epsilon^2 D_0$$

which implies $\mathbf{z} \in \mathcal{B}(\mathbf{z}^\diamond, c\epsilon\|\mathbf{z}^\diamond\|_2)$ for some constant $c$. A similar argument holds for $z_j^{(0)}$ and $\rho z_j^{\diamond,(0)}$ using (Sup.36). This implies the claim of the lemma.

We now proceed to prove our claim that either (Sup.35) or (Sup.36) holds. We make the following observations:

$$|\beta_j| \leq 1, \tag{Sup.37}$$

$$|\alpha_j| \leq \frac{1 + \sin\theta_j^{(0)}}{2\pi\|z_j^{(0)}\|_2} \tag{Sup.38}$$

for all $j$. Following similar arguments from [1, Proof of Lemma 8] and using (Sup.37) and (Sup.38), we can deduce that $M \leq 2$. Hand et.al. in [1] also establish using $M \leq 2$, (Sup.37) and (Sup.38) that for all $j$, $\theta_j^{(0)}$ is either small, i.e., $\theta_j^{(0)} \approx 0$, or large, i.e., $\theta_j^{(0)} \approx \pi$. We refer to [1, Proof of Lemma 8] for details. Now we consider those two cases.

**Small angle case:** Recall that we eventually aim to show either that (Sup.35) or (Sup.36) holds. Therefore under the assumption that $\theta_j^{(0)}$ is small, it is enough to show that $\|z_j^{(0)}\|_2 \simeq \|z_j^{\diamond,(0)}\|_2 = 1$. Assume that $\theta_j^{(0)} = \mathcal{O}(\epsilon)$. Then $\sin\theta_j^{(0)} = \mathcal{O}(\epsilon)$ and $\cos\theta_j^{(0)} = 1 + \mathcal{O}(\epsilon)$. Combining these with $M \leq 2$, (Sup.33), (Sup.37) and (Sup.38) yields $\|z_j^{(0)}\|_2 = 1 + \mathcal{O}(\epsilon)$ which is the desired result.

**Large angle case:** Assume that $\theta_j^{(0)} = \pi + \mathcal{O}(\epsilon)$. Similarly as before, it suffices to show that $\|z_j^{(0)}\|_2 \simeq \rho$ for some constant $\rho$. We have $\sin\theta_j^{(0)} = \mathcal{O}(\epsilon)$ and $\cos\theta_j^{(0)} = -1 + \mathcal{O}(\epsilon)$. Combining these with $M \leq 2$, (Sup.33), (Sup.37) and (Sup.38) yields $\|z_j^{(0)}\|_2 \simeq \frac{2}{5\pi} = \rho$. This completes the proof.

$\square$

In the remaining we make the following abbreviations: $v_{\mathbf{z}} = v_{\mathbf{z},\mathbf{z}^\diamond}$, $h_{\mathbf{z}} = h_{\mathbf{z},\mathbf{z}^\diamond}$, and $S_\epsilon = S_{\epsilon,\mathbf{z}^\diamond}$.

### Sup.2.2 Finishing the proof for Theorem 1

We first prove Equation (5) of Theorem 1, i.e., the existence of local maximum at the origin. We begin by providing a descent direction:

$$v_{\mathbf{z},\mathbf{z}^\diamond} = (\mathbf{W}_{2,+,\mathbf{z}}\mathbf{W}_{1,+,\mathbf{z}})^\top(\mathbf{W}_{2,+,\mathbf{z}}\mathbf{W}_{1,+,\mathbf{z}})\mathbf{z} - (\mathbf{W}_{2,+,\mathbf{z}}\mathbf{W}_{1,+,\mathbf{z}})^\top(\mathbf{W}_{2,+,\mathbf{z}^\diamond}\mathbf{W}_{1,+,\mathbf{z}^\diamond})\mathbf{z}^\diamond. \tag{Sup.39}$$

This expression is the gradient of $J$ where $J$ is differentiable. By using the definition of one-sided directional derivative $D_{\mathbf{z}}$ compute that

$$D_{\mathbf{z}}J(0) = -\langle\mathbf{W}_{2,+,\mathbf{z}}\mathbf{W}_{1,+,\mathbf{z}}\mathbf{z}, \mathbf{W}_{2,+,\mathbf{z}^\diamond}\mathbf{W}_{1,+,\mathbf{z}^\diamond}\mathbf{z}^\diamond\rangle \leq -\frac{1}{16\pi}\|\mathbf{z}\|_2\|\mathbf{z}^\diamond\|_2 < 0,$$

where the inequality follows from Lemma Sup.4 and the second inequality follows sice $\mathbf{z}, \mathbf{z}^\diamond \neq 0$.

Proof of Equation (4) is more involved. It mainly follows the arguments in [1](Section 6.1) adapted to our block structure of the weight matrices. One needs to eventually compute the directional

derivative $D_{v_{\mathbf{z},\mathbf{z}^\diamond}}$ and shows that it is negative when $\mathbf{z}$ is outside of two neighborhoods given in (4). Lemma Sup.8 establishes that $S_{4\epsilon}$ is a subset of two small neighbors around $\mathbf{z}^\diamond$ and $-\rho\mathbf{z}^\diamond$. Therefore, it is sufficient to show that for all $\mathbf{z}$ outside $S_{4\epsilon}$, the derivative along the descent direction $v_{\mathbf{z}}$ is always negative. By definition, the (unnormalized) one-sided directional derivative of $J(\mathbf{z})$ in the direction of $v_{\mathbf{z}}$ at $\mathbf{z}$ is

$$D_{v_{\mathbf{z}}} J(z) = \lim_{t \to 0+} \frac{J(\mathbf{z} + t v_{\mathbf{z}}) - J(\mathbf{z})}{t}.$$

Since the function $\sigma(\mathbf{W}_2 \sigma(\mathbf{W}_1 \mathbf{z}))$ is both continuous and piecewise linear, it follows that for any $\mathbf{z} \neq \mathbf{0}$ and $v_{\mathbf{z}} \neq \mathbf{0}$ there exists a sequence $\{\mathbf{z}_n\} \to \mathbf{z}$ such that $J$ is differentiable at each $\mathbf{z}_n$, and $D_{v_{\mathbf{z}}} J(\mathbf{z}) = \lim_{n \to \infty} \nabla J(\mathbf{z}_n) \cdot v_{\mathbf{z}}$. Since $\nabla J(\mathbf{z}_n) = v_{\mathbf{z}_n}$, it follows that

$$D_{v_{\mathbf{z}}} J(\mathbf{z}) = \lim_{n \to \infty} v_{\mathbf{z}_n} \cdot v_{\mathbf{z}}.$$

Now we bound the right-hand side from below

$$
\begin{aligned}
& v_{\mathbf{z}_n} \cdot v_{\mathbf{z}} \\
&= h_{\mathbf{z}_n} \cdot h_{\mathbf{z}} + (v_{\mathbf{z}_n} - h_{\mathbf{z}_n}) \cdot h_{\mathbf{z}} + \\
& \quad h_{\mathbf{z}_n} \cdot (v_{\mathbf{z}} - h_{\mathbf{z}}) + (v_{\mathbf{z}_n} - h_{\mathbf{z}_n}) \cdot (v_{\mathbf{z}} - h_{\mathbf{z}}) \\
&\geqslant h_{\mathbf{z}_n} \cdot h_{\mathbf{z}} - \|v_{\mathbf{z}_n} - h_{\mathbf{z}_n}\|_2 \|h_{\mathbf{z}_n}\|_2 - \\
& \quad \|h_{\mathbf{z}_n}\|_2 \|v_{\mathbf{z}} - h_{\mathbf{z}}\|_2 - \|v_{\mathbf{z}_n} - h_{\mathbf{z}_n}\|_2 \|v_{\mathbf{z}} - h_{\mathbf{z}}\|_2 \\
&\geqslant h_{\mathbf{z}_n} \cdot h_{\mathbf{z}} - \epsilon \max(\|\mathbf{z}_n\|_2 , \|\mathbf{z}^\diamond\|_2) \|h_{\mathbf{z}}\|_2 \\
& \quad - \epsilon \max(\|\mathbf{z}\|_2 , \|\mathbf{z}^\diamond\|_2) \|h_{\mathbf{z}_n}\|_2 \\
& \quad - \epsilon^2 \max(\|\mathbf{z}_n\|_2 , \|\mathbf{z}^\diamond\|_2) \max(\|\mathbf{z}\|_2 , \|\mathbf{z}^\diamond\|_2),
\end{aligned}
$$

where the first inequality follows from the triangle inequality and the second is a result of Lemma Sup.7.

As $h_{\mathbf{z}}$ is continuous in $\mathbf{z}$ for all nonzero $\mathbf{z}$, it follows that for any $\mathbf{z} \notin S_{4\epsilon}$,

$$
\begin{aligned}
\lim_{n \to \infty} v_{\mathbf{z}_n} \cdot v_{\mathbf{z}} &\geqslant \|h_{\mathbf{z}}\|_2^2 - 2\epsilon \|h_{\mathbf{z}}\|_2 \max(\|\mathbf{z}\|_2 , \|\mathbf{z}^\diamond\|_2) \\
& \quad - \epsilon^2 \max(\|\mathbf{z}\|_2 , \|\mathbf{z}^\diamond\|_2)^2 \\
&\geqslant \frac{\|h_{\mathbf{z}}\|_2}{2} \big[ \|h_{\mathbf{z}}\|_2 - 4\epsilon \max(\|\mathbf{z}\|_2 , \|\mathbf{z}^\diamond\|_2) \big] \\
& \quad + \frac{1}{2} \big[ \|h_{\mathbf{z}}\|_2^2 - 2\epsilon^2 \max(\|\mathbf{z}\|_2 , \|\mathbf{z}^\diamond\|_2)^2 \big] \\
&> 0
\end{aligned}
$$

The second inequality uses the definition of $S_{4\epsilon}$. Consequently, $D_{v_{\mathbf{z}}} J(\mathbf{z}) = \lim_{n \to \infty} v_{\mathbf{z}_n} \cdot v_{\mathbf{z}} > 0$, and thus $D_{-v_{\mathbf{z}}} J(\mathbf{z}) < 0$ for all $\mathbf{z} \notin S_{4\epsilon}$. Proof is finished by applying Lemma Sup.8.

$\square$

## Footnotes

*Both authors contributed equally to this work. Ulas Ayaz is presently affiliated with Lyft, Inc.