[Reviews · NeurIPS 2018]

Reviewer 1



This paper proves the invertibility of two-layer convolutional generative networks when partial measurements are given, under certain conditions: (i) weights are Gaussian or sub-Gaussian; (ii) the activation function is ReLU; (iii) the stride is the same as filter size. An empirical evidence, based in GAN training under MNIST and Celeb datasets, is provided to conjecture that the invertibility holds for more-than-2-layer convolution networks with different activation functions and strides sizes. Strength: S1: As indicated in a couple of places, the paper can provide a sufficient condition (stated in Thm1) for ensuring a mode collapsing problem, one of the critical problems in the GAN context. S2: Technical contribution looks non-trivial, based on the permutation technique. S3: The paper is very well-written and the proof is well streamlined. S4: An interesting empirical evidence is left for sparking further research directions. Weakness: W1: The proof is limited to the 2-layer network and s=filterSize case, while [9] (which forms the basis of the proof technique in this paper) deals with an arbitrary number of layers. At least the main difficulty that prevents extension if any should be explained in the paper, and any insight for progresses are desired to be included. W2: One of the interesting works is to identify the minimal sampling rate for perfect reconstruction, which is missing in the current work. W3: Comparison to [7] (which looks quite related) is not clearly described. Elaboration might help.

Reviewer 2



After reading the rebuttal, I favor acceptance but would strongly encourage the authors to carefully edit for clarity. As it is currently written, the paper is awash in undefined terms, poorly defined terms, and verbose notation. My list of notational comments is not intended to be an edit list. Some of the specific suggestions may well be incorrect, but my comments should hopefully direct your attention to portions of the document which need work. The paper presents a finding that while Convolutional GANs are theoretically hard to optimize due to non-convex loss surfaces, in practice, and with a few assumptions that do not seem to be too onerous, convolutional GANs can be minimized in their latent variable with respect to a partial output (sample). The result seems to largely be a result that strided convolutions can be written as a Toeplitz Matrix, which can be rewritten to be block diagonal with a permutation matrix, and then is provably invertible as long as the filter weights are gaussian or sub-gaussian. Positives: 1. The theoretical contributions seem to be mature 2. figure 4 a,b appear to validate the fact that 0 is a maximum, and -rho z and rho z are minima for a two layer relu network. 3. Results on "inpainting" images is currently interesting. A recent paper, published around the time of submission, Deep Image Prior Ulyanov et al CVPR 2018 fixes the latent variables and trains CNN parameters to recover an image. They show that doing this allows for lossy image recovery but do not provide any theoretical findings. Negatives: 1. While I think your proof is mature, I'm worried that it's too constrained. Two layer networks are only so interesting, and it's unclear to me how badly the loss surface under your analysis would blow up as more layers are added. 2. I'm not sure if this is a large issue for your proof, you seem to rely on the effective weights with relu taken into account being Gaussian, but your experimental validation of weights (sec 4.10) is on the raw weights. 3. Significant portions of the paper are difficult to follow due to typos or failure to define terms in the main text. (list below) footnote 2: "Collpase" line 95: "sub-sapling" line 105: "n2=n" should be n2=n1? line 111: "yAx" should be y=Ax? eq 3: this should be z {diamond} rather than z {hat} eq 4: this should be |Ay - AG(z)| right? You aren't trying to reconstruct the 0s section 2.1 you seem to be describing in detail the construction of a Toeplitz matrix, but never say so. eq 5: I don't think you define \mathit{B} in the main text.

Reviewer 3



This paper presents a new theoretical result on the invertibility of convolutional neural networks. The presented result is demonstrated (thoroughly in supplementary material) for a two-layer network, and claims are made about the empirical validity of the result on deeper networks. Invertibility is here intended as the recovery of latent code given partially observed network output (i.e. missing pixels), in the context of convolutional generative network. I think the direction of this paper's study is very interesting to better understand neural networks. The theoretical result presented shows that the mapping from the latent code space to the network output (i.e. the image space) is bijective. This result is demonstrated only for a two-layer convolutional generative network with ReLU, while the general case is let as future work. Could you clarify the choice of a two-layer network? Is a two-layer network more tractable/significantly easier to analyze with respect to, for example, a three-layer network? What is the main difficulty in proving a more general result on deeper networks? Line 182 of the paper says: "This proof can be extended to deeper networks with the same machinery." Could you clarify why you can only provide empirical evidence for the deeper network case? Why was the general result difficult to include in the paper? And if there is a specific difficulty, how would you suggest to address it? The same reasoning applies to the different activation functions: what was the main challenge in proving the results for different activation functions, and how would you address it? I think it would be important to clarify these points in the paper. The proof of the main result mostly follows arguments in the archival paper “Global Guarantees for Enforcing Deep Generative Priors by Empirical Risk” by Hand and Voroninski (cited), with some extensions to deal with the permutation applied to the weight matrices. While it seems that not much literature tackles this invertibility problem from the same angle, the cited “Towards Understanding the Invertibility of Convolutional Neural Networks” by Gilbert et al. (2017) is a relevant study, as well as other compressive sensing methods (e.g. the two cited works [4,5]). How does the proposed approach compare with those works? A quantitative evaluation of the proposed solution, and comparative experiments with the other relevant works are missing. I would like to see a quantitative evaluation of the proposed solution on the experiments presented. I would also like to see how the proposed solution compares with other relevant approaches. The paper is generally well written and pretty readable. Some claims seem generic, e.g. “we exploit the hierarchical nature of images and other natural signals by leveraging the powerful representation capability of deep learning.” (lines 63-65). Could you clarify how this is supported by the rest of the paper? For example, only a two-layer network is considered for the proof of the main result, and no guarantees are provided for the validity of the result for deeper networks apart from empirical results (which are, in my opinion, not sufficient to support this claim). Could you clarify how the experiments presented allow to see that the latent code was successfully recovered? I can only see reconstruction results, but there is no direct comparison between the “actual” latent code and the recovered one (which seems to be the point of this study). One of the assumptions of the proposed approach is the Gaussian weight assumption. How restrictive is this assumption in practice? Could you provide examples where this assumption would not hold? It is claimed that the proposed result "provides a sufficient condition to avoid mode collapses" (line 195). However this claim is not supported by experimental results. I would like to see, for example, an experiment where the mode collapse happens if the proposed solution is not applied. Could you clarify how the permutation applied to the weight matrices affects the geometry of the latent space? How does this operation effectively act on the latent code representation? Can the permutation operation be interpreted as a way to disentangle features in the latent code? Are there information losses or computational costs associated to it? After reading the author rebuttal, I tend to vote for the acceptance of this paper.